# Influence of Fiber Type and Dosage on Tensile Property of Asphalt Mixture Using Direct Tensile Test

**DOI:** 10.3390/ma16020822

**Published:** 2023-01-14

**Authors:** Shuyao Yang, Zhigang Zhou, Kai Li

**Affiliations:** Key Laboratory of Road Structure and Material Ministry of Communication, Changsha University of Science & Technology, Changsha 410114, China

**Keywords:** asphalt mixture, granular lignin fiber, flocculent lignin fiber, basalt fiber, fiber dosage, monotonic tensile tests, strain-controlled direct tensile fatigue tests, SEM

## Abstract

In engineering practice, fiber addition is a frequently used method to improve the tensile property of asphalt mixture. However, the optimum fiber type and dosage have not been determined by direct tensile tests. In this paper, monotonic tensile tests were conducted on three kinds of stone mastic asphalt (SMA13) mixtures, that is, granular-lignin-fiber-reinforced SMA (GFSMA), flocculent-lignin-fiber-reinforced SMA (FFSMA), and basalt-fiber-reinforced SMA (BFSMA) at different fiber dosages to probe the influence of fiber dosage on their tensile mechanical indexes (tensile strength, ultimate strain, elastic modulus, and strain energy density) and to determine the optimum dosage of each kind of fiber. The results showed that with the elevation of fiber dosage, the tensile strength, elastic modulus, and strain energy density of all three kinds of asphalt mixtures increased first and then decreased, while the ultimate strain increased constantly. The optimum dosage was 0.50 wt%, 0.45 wt%, and 0.50 wt% for granular lignin fiber, flocculent lignin fiber, and basalt fiber, respectively. On this basis, strain-controlled direct tensile fatigue tests were conducted on the three kinds of asphalt mixtures at the corresponding optimum fiber dosage. The results indicated that asphalt mixture reinforced with 0.50 wt% granular lignin fiber exhibited ideal direct tensile fatigue performance with respect to fatigue life and accumulative dissipated energy. Therefore, granular lignin fiber is recommended as the favorable fiber type, and its optimum dosage is 0.50 wt%. Moreover, scanning electron microscopy (SEM) demonstrated that the essence of the impact of fiber dosage and type on the tensile property of SMA is whether the reinforcement effect on the mixture matrix outweighs the negative effect of the defects between fiber and mixture matrix, or whether the reverse applies.

## 1. Introduction

Asphalt concrete pavement has been widely used in highway construction by virtue of driving comfort, low noise level, and simple maintenance. Even though asphalt mixture in the surface course, which is the main structural layer of the highway, is mainly intended for bearing pressure, studies on the mechanical properties and failure mechanism of asphalt mixture in tension are still very necessary, which can be beneficial for getting a comprehensive understanding of the performance of asphalt mixture. Moreover, in engineering practice, many defects have occurred on asphalt concrete pavement resulting from tensile performance deficiency, among which cracking is the most typical one [1,2]. Once cracking occurs, it can induce stress concentration in asphalt mixture, which can not only accelerate the failure of the pavement structure, but also affect driving comfort and even driving safety [3]. Therefore, it is of great significance to improve the tensile properties of asphalt mixture.

Various kinds of additives have been used to improve the tensile properties of asphalt mixtures [4,5,6]. Among them, fibers have gained the most extensive research attention. It has been demonstrated that fibers such as lignin fiber, basalt fiber, carbon fiber [7], steel fiber [8], glass fiber [9], and polyester fiber [10] can markedly enhance the tensile properties of asphalt mixtures. Lignin fiber and basalt fiber are the most commonly used types [11], and a lot of research is focusing on the comparison of their improvement effect on tensile properties to determine the ideal one. Guan et al. [12], Zhao et al. [13], Eisa et al. [14], and Gupta et al. [15] found that both the tensile strength and tensile strain of asphalt mixture reinforced by basalt fiber were higher than those of asphalt mixture reinforced by lignin fiber. Furthermore, Wu et al. [16] took the aging factor into account and found that the mixture reinforced with lignin fiber exhibited poorer tensile properties than the mixture reinforced with basalt fiber, no matter what the aging degree was. It was worth noting that the above literature showed that basalt fiber performed better in tensile property improvement than lignin fiber. In contrast, Zhu et al. [17] found that lignin fiber could improve tensile performance more remarkably than basalt fiber. Moreover, Zhang et al. [18] found that lignin fiber enhanced the overall performance of asphalt mixture more markedly than basalt fiber.

Besides the ideal fiber type, the optimum fiber dosage should also be determined. If the fiber dosage is too low, the fiber can hardly exert its reinforcement effect and will become a superfluous inclusion or defect. If the fiber dosage is too high, matrix discontinuity may occur, which can retard load transfer and even degrade the performance of the fiber [19,20]. It was revealed that the anti-tensile performance of asphalt mixture mainly depends on the bonding strength between asphalt mortar and aggregate [21]. The bonding strength is reported to be closely associated with basic indexes of asphalt mortar, such as viscosity and dynamic shear modulus, which can vary after fiber incorporation [22,23,24,25]. Therefore, fiber dosage is an important issue that needs to be taken into consideration for a favorable tensile performance.

Indoor evaluation methods for the tensile performance of asphalt mixture mainly include the indirect tension test [26,27,28], bending beam test [29,30], semi-circular bending test [23,31], and direct tension test [32]. Research by direct tension test is scarce, and studies on fiber-reinforced asphalt mixture by direct tension test are even fewer. Additionally, it is believed that the regularity and accuracy of the strain and stiffness modulus measured by indirect tension tests are not satisfactory, and the stress mode of the specimen differs greatly from the actual stress state of the actual structure. The test results of bending beam tests lack stability and the manufacture craft of the specimen for semi-circular bending tests is complex [33]. For direct tensile tests, the specimen manufacture craft is easier, the loading mold is simpler, and the direct acquisition of stress–strain data is easier; in addition, the specimen is less discrete and more sensitive to the test conditions and material parameters and can better reflect the actual stress state of the structure. These advantages render the direct tensile test the ideal one among these evaluation methods [34].

In view of this, in this paper, stone mastic asphalt (SMA) mixtures modified with granular lignin fiber (GFSMA), flocculent lignin fiber (FFSMA), and basalt fiber (BFSMA) were selected as the research object. The monotonic tensile test was carried out on the asphalt mixtures at different fiber dosages to analyze the influence of fiber dosage on the stress–strain curves and fracture morphology of tensile specimens. Furthermore, the effects of dosage on the tensile mechanical indexes (tensile strength, ultimate strain, elastic modulus, and strain energy density) of the three kinds of fiber-reinforced asphalt mixtures were explored, and the optimum dosage for each kind of fiber was determined accordingly. Then, the strain-controlled direct tensile fatigue test was carried out on the three kinds of fiber-reinforced asphalt mixtures at the corresponding optimum fiber dosage. Finally, the influence mechanism of fiber type and dosage on the tensile properties of asphalt mixture was analyzed by scanning electron microscopy (SEM). The research results can provide a basis for fiber selection for mixture design in engineering practice.

## 2. Materials and Methods

### 2.1. Materials

#### 2.1.1. Asphalt

Styrene-butadiene-styrene (SBS)-modified asphalt (PG 76-22) used in this research was provided by Shell (Xingyue) Co. Ltd. (Foshan, China). Their basic performance indexes are shown in Table 1.

#### 2.1.2. Fiber

Granular lignin fiber, flocculent lignin fiber, and basalt fiber used in this study were provided by Linxiang Building Materials Co. Ltd. (Changsha, China). Their basic performance indicators (provided by the manufacturer) are shown in Table 2, and the macro morphologies are displayed in Figure 1. It is stipulated in Technical Specification for Construction of Highway Asphalt Pavements (JTG F40-2004) that the dosage of lignin fiber should not be less than 0.30 wt% of the total weight of the mixture, and the dosage of basalt fiber should not be less than 0.40 wt%. Accordingly, the dosage of three kinds of fibers was set as follows: granular and flocculent lignin fibers (0.30 wt%, 0.40 wt%, 0.45 wt%, 0.50 wt%, and 0.55 wt%); basalt fiber (0.40 wt%, 0.45 wt%, 0.5 wt%, 0.55 wt%).

#### 2.1.3. Aggregates

Diabase was selected as coarse aggregate, limestone as fine aggregate, and limestone powder as mineral powder. Their basic technical performance indexes are presented in Table 3.

#### 2.1.4. Gradation

SMA13 gradation was used in this study and the gradation composition is presented in Table 4.

#### 2.1.5. Optimum Asphalt Content (OAC)

The follow-up experiments were carried out on asphalt mixtures reinforced with three kinds of fibers at the corresponding optimum asphalt content (OAC), which had been obtained through the Marshall test, and the OAC results are displayed in Table 5. From Table 5, it could be found that OAC value increases with the adding of fiber dosage for each kind of fiber-reinforced asphalt mixture and the growth gradient all tended to be diminished with the elevation of fiber dosage. Among them, the growth gradient for GFSMA was the most notable, followed by BFSMA, and FFSMA was the last. Moreover, for mixtures at the same fiber dosage, the OAC value of FFSMA was the largest, followed by GFSMA, and BFSMA was the last. This might be attributed to the fact that lignin fiber has a larger specific surface area, thereby showing more excellence in absorbing asphalt than basalt fiber.

### 2.2. Specimen Preparation and Test Protocol

#### 2.2.1. Mixing

The dry-mixing technique was adopted in this study to disperse the flocculent and bunched fibers, which was made easier by the friction of the aggregate [35]. The specific procedure is as follows: First, the fiber and aggregate were mixed together at 180 °C for 60 s, then asphalt was added and mixed for 90 s, and finally, the powder was added and mixed for 90 s.

#### 2.2.2. Monotonic Tensile Test

The main purpose of monotonic tensile tests was to explore the influence of fiber dosage on the tensile property indexes of three kinds of fiber-reinforced asphalt mixtures so as to determine the optimum dosage for each kind of fiber.

Considering the excellent consistency with the field conditions, the wheel rolling compaction method was adopted in this paper to mold the rutting plates of asphalt mixture (300 mm × 300 mm × 50 mm) according to the Standard Test Method of Bitumen and Bituminous Mixtures for Highway Engineering (JTG E20-2011). Then the rutting plates were sawed into trabecular specimens (250 mm × 50 mm × 50 mm) along the wheel trace direction by the single-sided saw.

MTS 810 testing machine manufactured by MTS Corporation (Eden Prairie, MN, USA) was applied in the test. Each of the two ends of the trabecular specimens was stuck to a steel plate using epoxy resin steel glue. After 48 h, when the strength of the steel glue was built, specimens were placed in the assorted environment chamber with a target temperature of 20 °C for 4 h, and then they were taken out and clamped on the instrument for tests. In order to eliminate the eccentric tension in the loading process, pre-tensioning was carried out before each test. The loading speed was controlled at a displacement control mode by 5 mm/min [36]. The force value was acquired by the computer system of the loading system, and the strain value was obtained by a pair of extensometers symmetrically arranged on the two cutting sides of each trabecular along the longitudinal direction. In this test, four evaluation indexes were used, that is, the tensile strength, the ultimate strain, elastic modulus, and the strain energy density, and they were defined in Equations (1)–(4), respectively.
(1)$$\sigma_{max}=\frac{F_{max}}{bd}$$
(2)$$\varepsilon_{max}=\frac{\varepsilon_{1max}+\varepsilon_{2max}}{2}$$
(3)$$S=\frac{\sum_{i=1}^{n}\left(\varepsilon_{i}-\frac{\sum_{i=1}^{n}\varepsilon_{i}}{n}\right)\left(\sigma_{i}-\frac{\sum_{i=1}^{n}\sigma_{i}}{n}\right)}{{\left(\varepsilon_{i}-\frac{\sum_{i=1}^{n}\varepsilon_{i}}{n}\right)}^{2}}$$
(4)$$\omega =\int_{0}^{\varepsilon_{max}}\sigma d\varepsilon$$
where $F_{max}$ = peak loading value in the course of tension; b = width of the specimen (50 mm); d = height of the specimen (50 mm); $\varepsilon_{1max}$ = strain value measured by one of a pair of extensometers when loading attains the peak; $\varepsilon_{2max}$ = strain value measured by the other of a pair of extensometers when loading attains the peak; $\left(\;\varepsilon_{i},\sigma_{i}\right)=\mathrm{data}\;\mathrm{of}\;\mathrm{the}\;\mathrm{elastic}\;\mathrm{stage}\;\mathrm{of}\;\mathrm{the}\;\mathrm{stress}-\mathrm{strain}\;\mathrm{curves}$; ∫ = integral of the stress-strain envelope area; σ = stress in the tension process; and ε = strain in the tension process.

#### 2.2.3. Direct Tensile Fatigue Test

The direct tensile fatigue test was performed to determine an ideal fiber type that can most remarkably improve tensile fatigue. The procedure (preparation, clamping, and preservation) was the same as that for monotonic direct tensile tests. The strain-controlled mode, under which the stress–strain situation of asphalt mixtures was more consistent with that of actual pavements, was selected as the cyclic control mode. The procedure for determining the strain levels was as follows: firstly, the average of the ultimate strain of three kinds of fiber-reinforced asphalt mixtures at the corresponding optimum fiber dosage was calculated; then, the average value could be multiplied by 0.4, 0.5, 0.6, and 0.7 to acquire the target controlled strain. The half-sine waveform at a frequency of 10 Hz was applied. The test temperature was 20 °C. The fatigue life was defined as the cycle number at which the stiffness modulus drops to 50% of that of the 100th cycle (named as initial stiffness modulus).

The fatigue process of asphalt mixtures could be understood as the resistance of their internal structure to the impact of external action, in which energy would be continuously consumed for internal reconstruction. The accumulative dissipated energy was used as an index to evaluate the fatigue resistance besides the aforementioned index N, and it can be calculated by Equation (5):(5)$$W=\sum_{I=1}^{N}\pi \sigma_{i}\varepsilon_{i}sin(2\pi f\Delta t)$$
where *W* = accumulative dissipated energy; *f* = the applied loading frequency (Hz); and ∆*t* = the time of strain peak lagging behind the stress peak (s).

#### 2.2.4. SEM Test

For a better understanding of the impact mechanism of fiber dosage and type on the tensile properties of asphalt mixtures, samples sliced from the fracture section of monotonic tensile specimens were observed by scanning electron microscope. In the test, each sample was plated with a layer of 20 mm gold powder in a vacuum and then put into the electron microscope sample room for observation.

## 3. Discussion

### 3.1. Results of Monotonic Tensile Test

#### 3.1.1. Tensile Stress–Strain Curves

The stress–strain curves of GFSMA, FFSMA, and BFSMA were depicted in Figure 2. On the whole, all of those stress curves could be divided into the following three stages: elastic stage, strain hardening stage, strain softening stage. (1)Elastic stage: The stress–strain curve in this stage was close to a straight line. Specifically, when the strain was within 10–30% of the ultimate strain, the stress showed a synchronous linear upward trend with the elevation of strain.(2)Strain hardening stage: The stress–strain curve in this stage presented an obvious deviation from its initial slope and was similar to an arc. During this stage, stress and strain both kept growing but the growth rate of the former was lower than the latter. By contrast, the growth rate of stress slowed down compared to that of stress with the previous stage.(3)Strain softening stage: When stress reached the peak value, strain continued increasing while stress began to gradually decease. When the stress reaches the peak stress, the strain continues to increase while the stress begins to gradually decrease. This phenomenon might be explained as the following: When the stress and strain inside the specimen reached a certain extent, some fibers could be pulled out from the crack or even pulled off so that the resistance of the specimen to external loading was weakened and the tensile strength dropped eventually.

In addition, for all the three kinds of fiber-reinforced asphalt mixtures, the variation of fiber dosage had an impact on their own stress–stain curves to some degree. As fiber dosage increased, the wholeness of all of the curves tended to be plumper; the vertical span of the curves firstly went rising and then moved down; both the length and the slope of the first stage of the curves climbed up first and then declined; the arc coverage span of the second stage of the curves was first elongated and then shortened while the arc curvature exhibited the reverse variety; and the falling gradient of the third stage of the curves tended to be flattened.

#### 3.1.2. Feature Analysis on Fracture Section

Given space constraints, the effects of fiber dosage on the crack morphology and fracture surface morphology of specimens were illustrated with GFSMA as an example. The results are presented in Figure 3 and Figure 4, respectively.

From Figure 3, it was observed that when the dosage was 0.3 wt%, the fractured section was generally parallel to the transverse section of the specimen. The cracks of the specimen were generally linear and did not yet extend longitudinally to the specimen. However, the edges of the section showed slightly uneven features, which were caused by the bridging effect of fibers. It could be observed from Figure 3 that there was some matrix debris on the fracture surface, and the fracture surface was slightly undulating. As the dosage continued climbing up, the fracture toughness of the specimen was significantly enhanced, and the whole crack or a segment thereof on the fracture surface was no longer flat. It could be easily noticed that cracks winded along the longitudinal direction of the specimen, and the crack width increased to a certain extent; in this case, the matrix may be torn. Some of the matrix was loosened and peeled off at the edge of the crack or even in the middle of the section during the test procedure. As a result, the fracture surface was very uneven, with matrix debris and fibers, and the two sections of the damaged specimens could not even be completely matched. However, when the dosage leaped to 0.55 wt%, the sinuousness of the macro cracks seemed to be less obvious than that of the mixture with 0.50 wt% fiber. Overall, from the perspective of the feature of the fracture section, the asphalt mixture at the fiber dosage of 0.50 wt% embodied higher fracture toughness than that at the other four dosages, which might cause it to have more excellent tensile properties.

In addition, comparing the five pictures in Figure 4, it was worth noting that there were more fibers on the fracture surface in Figure 3d compared with the other four pictures. In addition, as depicted by Figure 4d, some of the fibers had been pulled out of the matrix entirely and some had been broken. The two kinds of behaviors mentioned above both needed extra absorption of energy so that asphalt mixture exhibited more remarkable fracture toughness characteristics.

#### 3.1.3. Results of Tensile Strength

The tensile strength results of GFSMA, FFSMA, and BFSMA at different fiber dosages are presented in Figure 5. It can be seen that for all three types of asphalt mixtures, the addition of an appropriate dosage of fiber could improve their tensile strength, but when the fiber dosage exceeded a certain critical value, the tensile strength would decrease with the growth of the fiber dosage. The reason for the above phenomenon may be as follows: At an appropriate elevation of fiber dosage, the average fiber distance could be shortened so that there could be more fibers participating in the adhesion with matrix, which would strengthen the interface between the matrix and fiber and eventually improve the tensile strength of mixture. However, the continuous rising of fiber dosage brought about the enlargement of coverage of the fiber and then the narrowness of the effective bonding interface between fiber and asphalt, which could lead to the decline of tensile strength of the mixture instead.

The tensile strength of GFSMA at the fiber dosage of 0.40 wt% increased by 15% compared with that at the fiber dosage of 0.30 wt%. When the dosage reached up to 0.45 wt%, the tensile strength reached its peak value (about 0.42 MPa), which increased by 27% compared with that at the fiber dosage of 0.30 wt%. When the dosage continued increasing, the tensile strength began to decrease lightly but that at the fiber dosage of 0.55 wt% was still slightly higher than that at 0.30 wt% by 12%.

The tensile strength of FFSMA reached the peak value (about 0.55 MPa) at the fiber dosage of 0.40 wt%, which was about 47% higher than that at the fiber dosage of 0.30 wt%. When the dosage increased from 0.40 wt% to 0.45 wt%, the tensile strength tended to descend dramatically but was still mildly higher than that at the fiber dosage of 0.30 wt% by nearly 26%. When the dosage reached 0.50 wt%, the tensile strength dropped to a value that was almost equal to that at 0.30 wt%. Then, the tensile strength declined to 0.33 MPa but the decrease gradient slowed down when the dosage ascended to 0.55 wt%.

For BFSMA, the tensile strength reached the peak value (about 0.41 MPa) when the dosage attained 0.45 wt%, which was remarkably higher than that at the fiber dosage of 0.4 wt% by about 24%. With the dosage further increasing, however, the tensile strength showed a declining tendency. When the fiber dosage attained 0.55 wt%, the tensile strength was even below that at the fiber dosage of 0.4 wt% by 6%.

The fractional function was imported to execute fitting between the tensile strength and fiber dosage for the three kinds of fiber-reinforced asphalt mixtures. The results are shown in Figure 6. It can be seen that the fitting correlation coefficients R^2^ for the three kinds of fiber-reinforced asphalt mixtures ranged from 0.82 to 0.98, indicating that the fractional function could accurately describe the relationship between the tensile strength of the asphalt mixture and fiber dosage within 0.30–0.55 wt%.

#### 3.1.4. Results of Ultimate Strain

The ultimate strain results of GFSMA, FFSMA, and BFSMA at different fiber dosages are shown in Figure 7. From Figure 7, it can be seen that the addition of an appropriate amount of fiber based on the dosage stipulated in the specification could enhance the ultimate strain of the asphalt mixtures. This can be explained as follows: On one hand, fibers mentioned in this research had excellent resistance to tensile deformation; on the other hand, they could function as bridge in the mixture, which would share and transfer deformation form matrix. The above two factors both contributed to the improvement on mixture’s ultimate strain. In addition, with the increase of fiber dosage, more and more fibers were involved in the improvement issue, and then the ultimate stain exhibited a growing trend.

The ultimate strain of GFSMA at the fiber dosage of 0.4 wt%, 0.45 wt%, 0.50 wt%, and 0.55 wt% increased by 10%, 23%, 28%, and 48%, respectively, when compared with that at the fiber dosage of 0.30 wt%. In addition, the increment gradient at 0.55 wt% was the most significant.

For FFSMA, when the dosage was 0.40 wt%, the increment of the ultimate strain was very limited (only 3%) compared to the ultimate strain at 0.30 wt%, but with the continuous climbing of the dosage, the fiber improvement impact on the ultimate strain was more and more significant. Compared to the ultimate strain at 0.30 wt%, the ultimate strain increased by 17%, 28%, and 37%, respectively, for 0.45 wt%, 0.50 wt%, and 0.55 wt%. In addition, it was not difficult to find that with the increase in fiber dosage, the growth gradient of the ultimate strain tended to be flatter.

For BFSMA, as the fiber dosage went up from 0.40 wt% to 0.45 wt%, the enhancement effect of basalt fiber on the ultimate strain was not remarkable, with an increment of only 10%. When the dosage increased to 0.50 wt%, the ultimate strain increased sharply and was more than half of that at 0.40 wt%. When the dosage reached 0.55 wt%, the ultimate strain increased by 60% compared with that at 0.4 wt%. When the dosage increased to 0.50 wt%, the ultimate strain increased dramatically and was more than half of that at 0.4 wt%. When the dosage reached 0.55 wt%, the ultimate strain increased by 60% compared with that at 0.40 wt%, but the increase slowed down.

The ultimate strain and fiber dosage of the three kinds of fiber-reinforced asphalt mixtures were fitted, and the fitting curves are shown in Figure 8. The fitting results indicated that the first-order function could better describe the relationship between the tensile strength and the fiber dosage with the fitting correlation coefficients R^2^ in the range of 0.90–0.93 for all three kinds of fiber-reinforced asphalt mixtures.

#### 3.1.5. Results of Elastic Modulus

The elastic modulus of GFSMA, FFSMA, and BFSMA at different fiber dosages are presented in Figure 9. It can be seen that for GFSMA and FFSMA, the addition of an appropriate dosage of fiber could improve their elastic modulus, but when the fiber dosage exceeded a certain critical value, it would decrease with the increase of the fiber dosage. However, for BFSMA, the elastic modulus kept a gradual downward trend with the elevation of fiber dosage. The phenomenon might be explained by the difference between the variation gradient of the tensile strength and the ultimate strain.

For GFSMA, there was a marked growth of nearly 60% in the elastic modulus at the fiber dosage of 0.40 wt% compared with that at the fiber dosage of 0.30 wt%. Then, the elastic modulus kept a falling tendency at a stable descending gradient. However, it was worth noting that the elastic modulus was not below that at the fiber dosage of 0.3 wt% until the fiber dosage attained 0.55 wt%.

Similarly, the elastic modulus of FFSMA showed a slight increase by about 12% at the fiber dosage of 0.40 wt% than that at the fiber dosage of 0.30 wt%. When the dosage increased from 0.40 wt% to 0.45 wt%, the elastic modulus tended to descend. When the dosage reached 0.55 wt%, the elastic modulus dropped more sharply with a reduction rate of 66% compared with that at the fiber dosage of 0.3 wt%.

For BFSMA, as the fiber dosage increased from 0.40 wt% to 0.45 wt%, there was a growth by about 16% likewise. When the fiber dosage exceeded 0.45 wt%, the elastic modulus began to decline steadily and the elastic modulus at the fiber dosage of 0.50 wt% and 0.55 wt% descended by about 29% and 52%, respectively, compared with that at 0.4 wt%.

The fractional function was also imported to execute fitting between the elastic modulus and fiber dosage for the three kinds of fiber-reinforced asphalt mixtures. The results are shown in Figure 10. It can be seen that the fitting correlation coefficients R^2^ for the three kinds of fiber-reinforced asphalt mixtures ranged from 0.86 to 0.94, indicating that the fractional function could accurately describe the relationship between the elastic modulus of asphalt mixture and fiber dosage within 0.3 wt%–0.5 wt%.

#### 3.1.6. Results of Strain Energy Density

From the above analysis of tensile strength and ultimate strain, it could be summarized that the variation regulation for the tensile strength and the ultimate strain with the variation of fiber dosage for each kind of fiber-reinforced asphalt mixture showed great difference. In addition, the two indexes reached their own peak value at different fiber dosages, which could subsequently result in an inconsistency between the optimum fiber dosage determined, respectively, based on the two indicators. Therefore, a comprehensive indicator called strain energy density, which can reflect both the tensile strength and ultimate strain [37], was chosen to evaluate the tensile properties of the asphalt mixtures.

The results of the strain energy density of GFSMA, FFSMA, and BFSMA at different dosages are displayed in Figure 11. It can be seen that the strain energy density of all three types of fiber-reinforced asphalt mixtures ascended first and then descended, which was similar to the trend of the tensile strength. It can be illustrated as follows: Fibers were distributed irregularly and overlapped with each other to form a three-dimensional network, which could have “reinforcement” and “anti-cracking” effects. Moreover, the realization of the two effects mentioned above needed strain energy absorbed by energy. Therefore, with the increase of fiber dosage, there were supposed to be more strain energy in the mixture. However, at the same time, the continuous addition of fiber could lead to uneven distribution and even agglomeration, both of which could diminish the effect of “reinforcement” and “anti-cracking” of fiber and result in the drop of strain energy density in the end.

The strain energy density of GFSMA at 0.40 wt% and 0.45 wt% increased by 29% and 57%, respectively, compared with that at 0.30 wt%. When the dosage attained 0.50 wt%, the strain energy density reached the peak value (about 4481 MJ/m^3^), which was 67% higher than that at 0.30 wt%. When the dosage increased up to 0.55 wt%, the strain energy density was slightly lower than that at 0.50 wt% but was still more than half of that at 0.30 wt%.

The strain energy density of FFSMA at 0.4 wt% significantly increased by 76% compared with that at 0.3 wt%. When the dosage reached 0.45 wt%, the strain energy density reached the peak value (4439 MJ/m^3^), which was 87% higher than that at 0.30 wt%. With the continuous increase of dosage to 0.50 wt% and 0.55 wt%, although the strain energy density of FFSMA began to show a nearly uniform downward trend, it was approximately 56% and 27% higher than that at 0.30 wt%, respectively.

The strain energy density of BFSMA at 0.45 wt% leaped by 81% compared with that at 0.40 wt%. When the dosage went up to 0.50 wt%, the strain energy density reached the peak value (4112 MJ/m^3^), which was more than twice that at 0.40 wt%. When the dosage was 0.55 wt%, the strain energy density was nearly 1.2 times higher than that at 0.40 wt% but declined compared with that at 0.50 wt%.

The strain energy density of the three kinds of fiber-reinforced asphalt mixtures was fitted with fiber dosage, and the fitting image is displayed in Figure 12. The fitting results showed that the fractional function could describe the relationship between the strain energy density of all three kinds of asphalt mixtures and fiber dosage, and the fitting correlation coefficients R^2^ were in the range of 0.96–0.99.

Furthermore, it could be summarized from the above analysis results that within the given dosage range, the optimal dosage of granular lignin fiber, flocculent lignin fiber, and basalt fiber was 0.50 wt%, 0.45 wt%, and 0.5 wt%, respectively, based on the criteria of maximal strain energy density.

### 3.2. Results of Direct Tensile Fatigue Test

#### 3.2.1. Fatigue Life

The strain-controlled direct tensile fatigue tests were carried out for GFSMA, FFSMA, and BFSMA at the corresponding optimum dosage. The four controlled strain were 5048με, 6310με, 7572με, and 8834με. Five replicates were fabricated for each kind of fiber-reinforced asphalt mixture at each controlled strain. The fatigue life results are shown in Table 6. From Table 6, it could be concluded that within the given range of strain, the fatigue life of the three kinds of fiber-reinforced asphalt mixtures at the same strain was ranked as GFSMA > BFSMA > FFSMA. The fatigue life of GFSMA was 3.7 times and 2.2 times that of FFSMA and BFSMA, respectively, under the strain of 5048με; 4.6 times and 2.1 times longer under the strain of 6310με; 4.7 times and 1.7 times longer under the strain of 7572με; and 3.6 times and 1.8 times longer under the strain of 8834με.

In addition, it could also be seen that the fatigue life of the three types of fiber-reinforced asphalt mixtures exhibited varying degrees of decrease with the growth of strain level. Equation (6) was used to depict the relationship between fatigue life and strain level of three kinds of asphalt mixtures in double logarithmic co-ordinates. The fitting results are shown in Figure 13.

It could be found from Figure 13 that in terms of the *k* value, the three fiber-reinforced asphalt mixtures were ranked as BFSMA < FFSMA < GFSMA; in terms of the *c* value, the three asphalt mixtures were ranked as BFSMS < GFSMA < FFSMA. The above results showed that GFSMA had moderate sensitivity to strain and ideal fatigue resistance with the highest fatigue line position. By contrast, BFSMA had the lowest fatigue resistance and sensitivity to strain.

It could be found from the figure that the *k* value of the three fiber asphalt mixtures were in order as follows: BFSMA < FFSMA < GFSMA; the *c* value of the three fiber asphalt mixtures were in order as follows: BFSMS < GFSMA < FFSMA. The above results showed that GFSMA had moderate sensitivity to strain as well as the best fatigue resistance with the highest fatigue line position. By contrast, BFSMA was observed to be of the poorest fatigue resistance and sensitivity to strain.
(6)$$N=k{\left(\frac{1}{\varepsilon }\right)}^{c}$$
where *N* = fatigue life; *ε* = strain; *c* = sensitivity of fatigue life to the variation of strain (the slope of fitting curve); and *k* = fatigue resistance (the intercept of fitting curve).

#### 3.2.2. Cumulative Dissipated Energy

The cumulative dissipated energy results of three types of fiber-reinforced asphalt mixtures are shown in Figure 14. From Figure 14, it can be seen that in terms of the cumulative dissipated energy at the same strain, the three kinds of asphalt mixtures were ranked as GFSMA > BFSMA > FFSMA, which is consistent with the ranking in terms of fatigue life. Specifically speaking, the cumulative dissipated energy of GFSMA was 1.7 times and 1.2 times higher than that of FFSMA and BFSMA, respectively, under the strain of 5048με; 2.6 times and 1.5 times higher under the strain of 6310με; 2.5 times and 1.9 times higher under the strain of 7572με; and 2.4 times and 1.6 times higher under the strain of 8834με. It could be summarized that the tensile fatigue performance of GFSMA at 0.5 wt% is superior to that of FFSMA at 0.45 wt% or BFSMA at 0.50 wt% in terms of fatigue life and cumulative dissipative energy.

Based on the data of cumulative dissipated energy and fatigue life of three types of fiber-reinforced asphalt mixtures under each strain, fitting was performed in the double logarithmic co-ordinates using Equation (7) of the dissipated energy model proposed by Dijk and Visser. The fitting results (by 95% confidence limits) are shown in Table 7.
*W* = *AN^Z^*
(7)

where *W* = accumulative dissipated energy (MPa); *N* = fatigue life; and *A* and *z* = fitting parameter.

The relationship between each of the above two fitting parameters and strain can be approximately expressed by a linear function, and the fitting results are shown in Figure 15.

Finally, the relationship between the strain and the cumulative dissipative energy of the three fiber-reinforced asphalt mixtures could be obtained by plugging the relationship formula displayed in Figure 13 into Equation (7), and the derived formula is shown in Equation (8).
(8)$$W=10^{3.08\varepsilon -0.000671}N^{-0.31\epsilon +0.000137}$$

### 3.3. Analysis of SEM Test Results

The influence mechanism of fiber dosage on the tensile performance of asphalt mixture was explained by taking the cross-section morphology of a tensile specimen of GFSMA at five dosages as examples; the influence mechanism of fiber type on the tensile properties of asphalt mixture was explained by taking the cross-section morphology of tensile specimens of GFSMA, FFSMA, and BFSMA at the corresponding optimum dosage as examples. The SEM images acquired are shown in Figure 16.

Fibers are dispersed in the mortar in a disorderly overlapping manner and crisscross with the asphalt to form a three-dimensional network structure. The external load is generally directly applied to the matrix, and then the stress can be transferred to the fiber through the fiber end and the interface between the fiber and the matrix. The stress imposed on the fiber is rapidly diffused, which can prevent the crack from propagation. That is the very embodiment of the bridging effect of fiber.

Fibers can also absorb the asphalt to its surface. The roots of fibers are well-bonded with the asphalt to exert an anchoring effect, and the end forms an antenna with the asphalt to exert an interlocking effect [18]. With the addition of fiber to the mixture, the asphalt mortar film wrapped on the aggregate surface becomes thicker, impeding the mutual sliding between asphalt and aggregate, which will enhance the overall ability of the structure to resist external loads and then improve tensile properties. This reflects the reinforcement effect of fiber on the mixture matrix. Negative effects can be brought about once fiber dosage exceeds a certain limit.

Moreover, unevenness in the following aspects can be observed in Figure 16i,j. First, fibers are dispersed in the mixture unevenly. Second, fibers are mechanically twisted and knotted with each other in the local region. Third, the asphalt adsorbed on the fibers has an uneven “flocculent” distribution.

Furthermore, with the increase in the dosage, the wrinkle at the fiber anchorage becomes more and more prominent, which indicates the trend of stripping and pull-out from the “root”. From Figure 16j, it can be observed that there are pulled-out fibers, which can result in holes on the interface. The unevenness and pull-out of fiber can lead to undesirable negative effects, which will increase the internal defects of the mixture, degrade the internal continuity of the asphalt mixture, and reduce the compactness.

As long as the dosage of fiber is appropriate, the positive effect, namely, the bridging effect and adhesion effect, can be boosted. Nevertheless, when the dosage attains a certain value, the negative effect brought about by uneven dispersion and pull-out of fiber may outweigh the positive effect, which may thereby degrade the overall tensile property and crack the resistance of asphalt mixtures.

Based on the comparison among Figure 16g,h and Figure 16k–n, it can be found that basalt fiber has the best dispersion uniformity, followed by granular lignin fiber, and the dispersion uniformity of flocculent lignin fiber is the worst. This can be explained by the difference in the diameter of the three types of fibers. Specifically, a larger fiber diameter indicates a smaller number of fibers and a smaller amount of entanglement in the composited material [38]. Furthermore, basalt fiber has higher tensile strength and length–diameter ratio than the other two types of fibers, which contributes to its better performance in larger stress transfer and bridging. From this point of view, BFSMA is supposed to have better anti-tensile performance. However, what is noteworthy is that the asphalt film wrapped on the flocculent lignin fiber is the thickest, followed by the film wrapped on the granular lignin fiber, and the asphalt film wrapped on basalt fiber is the thinnest. The reason for this difference is that the lignin fiber has the largest specific surface area, as reflected by the largest number of convex structures observed on the surface, making it easier to absorb more and thicker asphalt. The superior tensile properties of GFSMA can be attributed to the situation where the positive effect brought about by the excellent adhesion properties outweighs its weakness in evenness and bridging.

## 4. Conclusions

In this research, the effects of fiber dosage and type on the tensile properties of three kinds of fiber-reinforced asphalt mixtures were investigated, respectively. In addition, the SEM test was performed to explore the influence mechanism of fiber dosage. According to the test results, the corresponding conclusions are summarized as follows:

The results of the monotonic direct tensile tests showed that when the fiber dosage keeps increasing within the given range, the tensile strength, elastic modulus, and strain energy density of the three kinds of fiber-reinforced asphalt mixtures first went up and then went down, while the ultimate strain showed a continuous rising trend.

In light of the criterion of maximal strain energy density, the optimum fiber dosage of GFSMA, FFSMA, and BFSMA was determined to be 0.50 wt%, 0.45 wt%, and 0.50 wt%, respectively. In addition, in terms of the strain energy density of three kinds of fiber-reinforced asphalt mixtures with the corresponding optimum dosage, the performance of GFSMA was the best, followed by FFSMA, and the performance of BFSMA was the worst.

The strain-controlled direct tensile fatigue tests showed that direct tensile fatigue of GFSMA at the fiber dosage of 0.50 wt% was superior to that of FFSMA at 0.45 wt% or BFSMA at 0.50 wt% in terms of fatigue life and cumulative dissipative energy. Therefore, it was concluded that granular fiber could more remarkably improve the tensile properties of asphalt mixtures, and its optimum dosage was 0.50 wt%.

The analysis results of the SEM test suggested that the essence of the influence of fiber dosage and type on the asphalt mixture is whether the reinforcement effect of the fiber on the mixture matrix outweighs the negative effect of interface defects between fiber and mixture matrix or the reverse applies.

## Figures and Tables

**Figure 1 materials-16-00822-f001:**
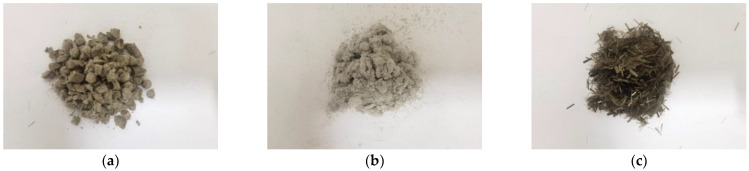
Macro morphology of fiber: (**a**) Granular lignin fiber; (**b**) Flocculent; (**c**) Basalt fiber.

**Figure 2 materials-16-00822-f002:**
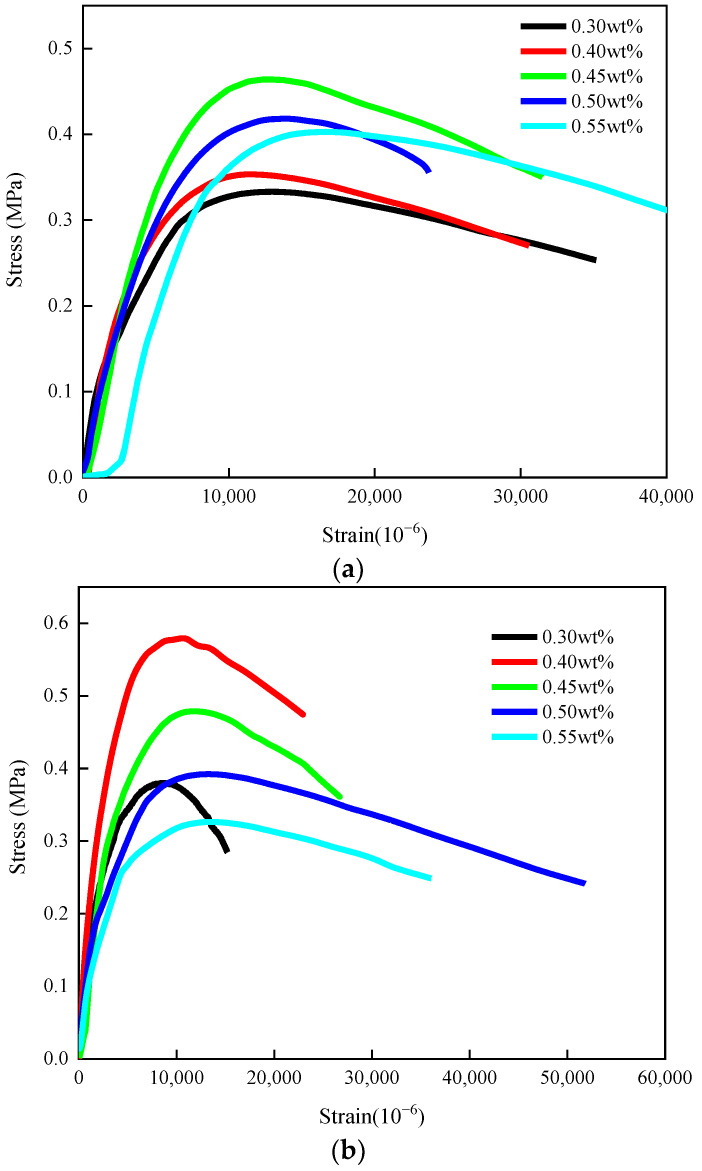
Stress–strain curves: (**a**) GFSMA; (**b**) FFSMA; (**c**) BFSMA.

**Figure 3 materials-16-00822-f003:**

Crack morphology of GFSMA specimens: (**a**) 0.3 wt; (**b**) 0.4 wt%; (**c**) 0.45 wt%; (**d**) 0.50 wt%; (**e**) 0.55 wt%.

**Figure 4 materials-16-00822-f004:**
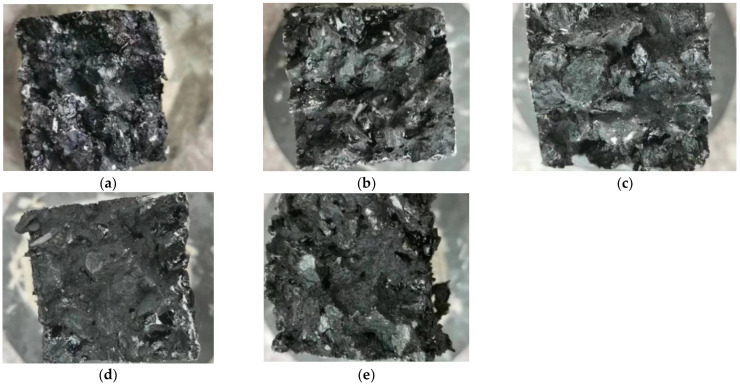
Fracture surface morphology of GFSMA specimens: (**a**) 0.3 wt%; (**b**) 0.4 wt%; (**c**) 0.45 wt%; (**d**) 0.50 wt%; (**e**) 0.55 wt%.

**Figure 5 materials-16-00822-f005:**
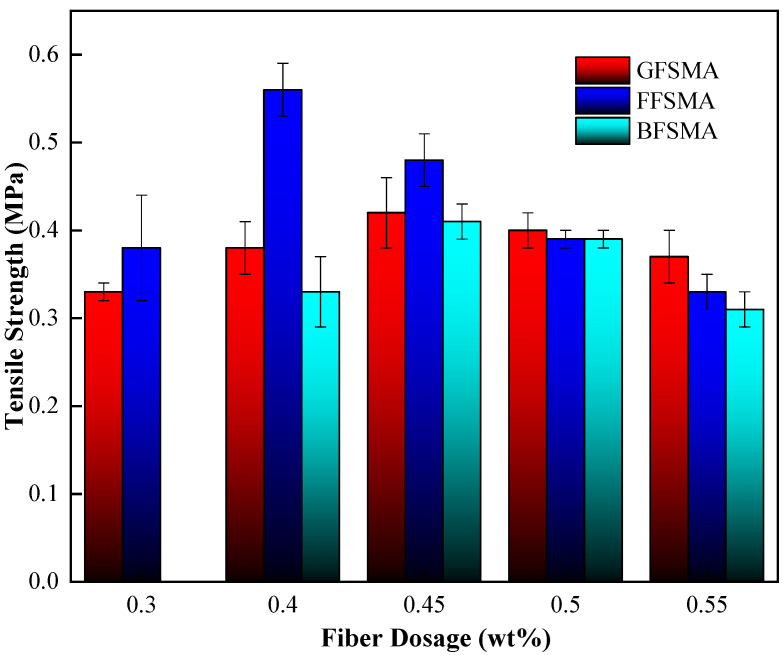
Tensile strength of mixtures.

**Figure 6 materials-16-00822-f006:**
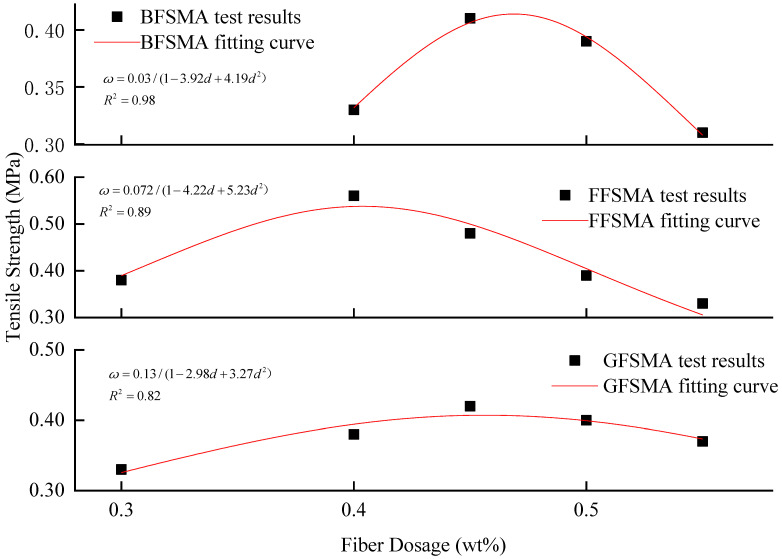
Fitting between fiber dosage and tensile strength.

**Figure 7 materials-16-00822-f007:**
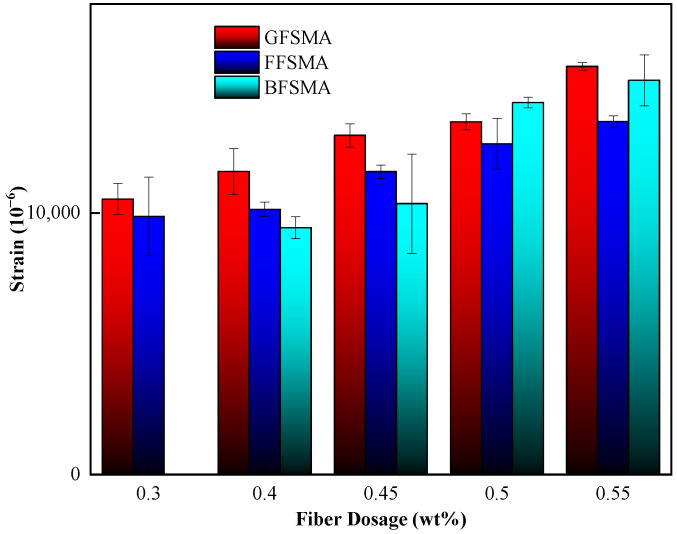
Ultimate strain of mixtures.

**Figure 8 materials-16-00822-f008:**
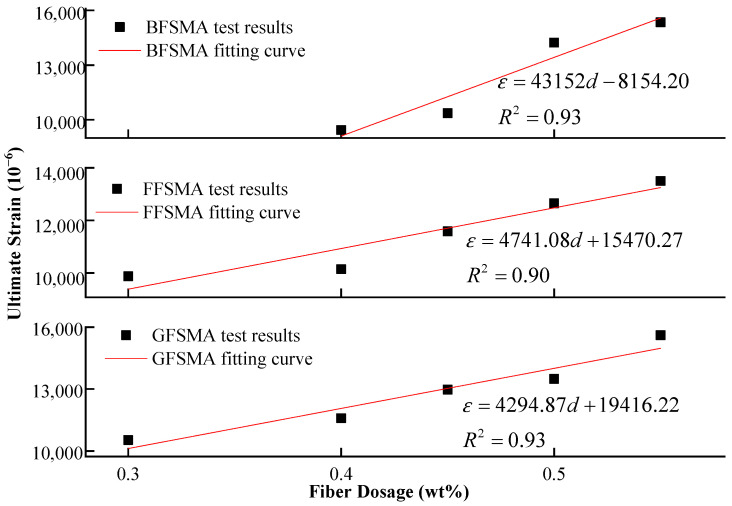
Fitting between fiber dosage and ultimate strain.

**Figure 9 materials-16-00822-f009:**
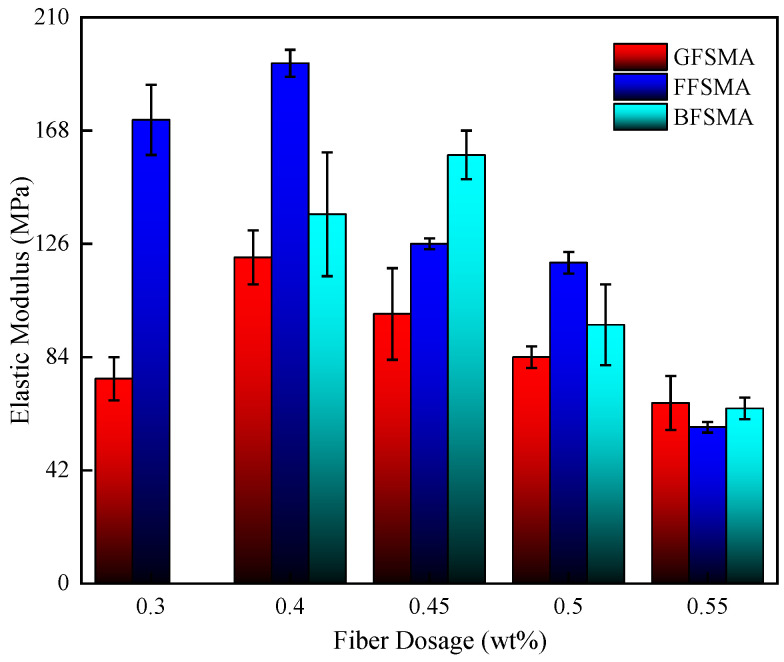
Elastic modulus of mixtures.

**Figure 10 materials-16-00822-f010:**
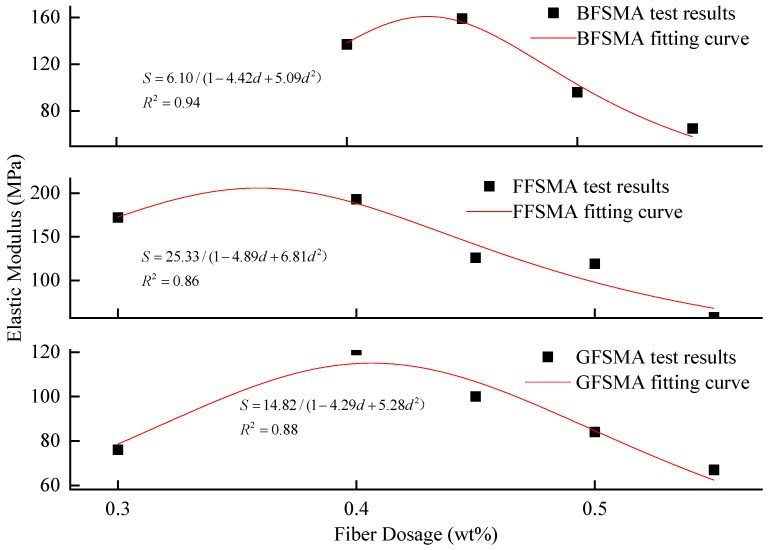
Fitting between fiber dosage and elastic modulus.

**Figure 11 materials-16-00822-f011:**
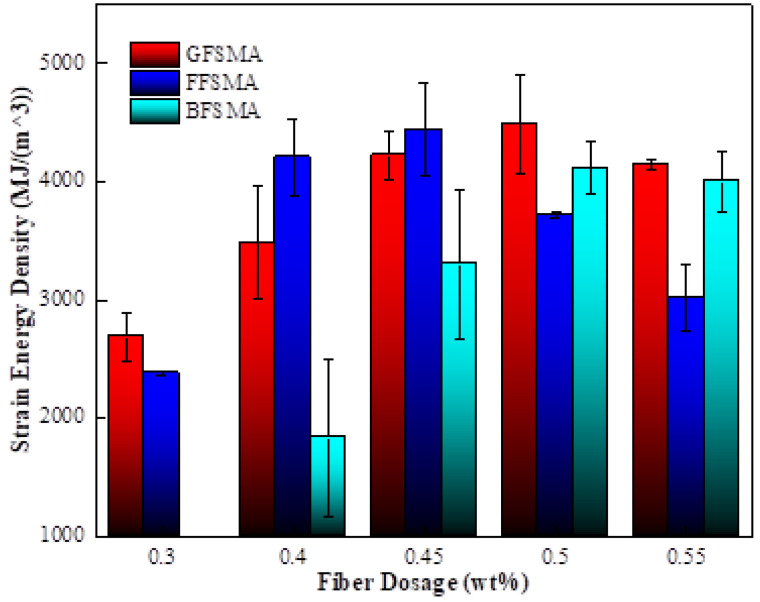
Strain energy density of mixtures.

**Figure 12 materials-16-00822-f012:**
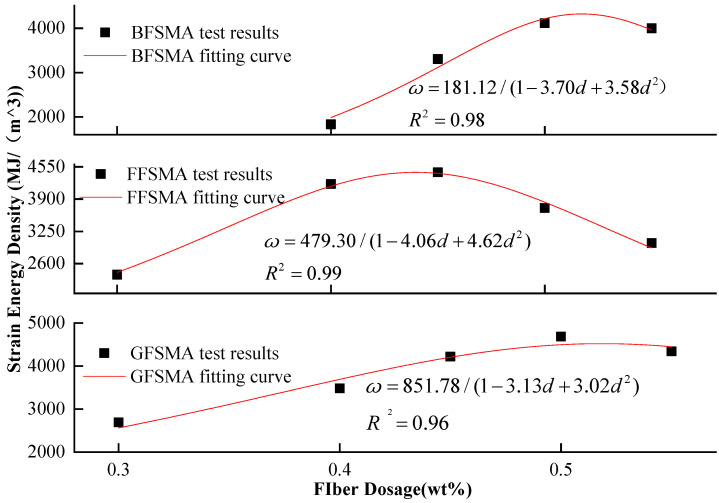
Fitting between fiber dosage and strain energy density.

**Figure 13 materials-16-00822-f013:**
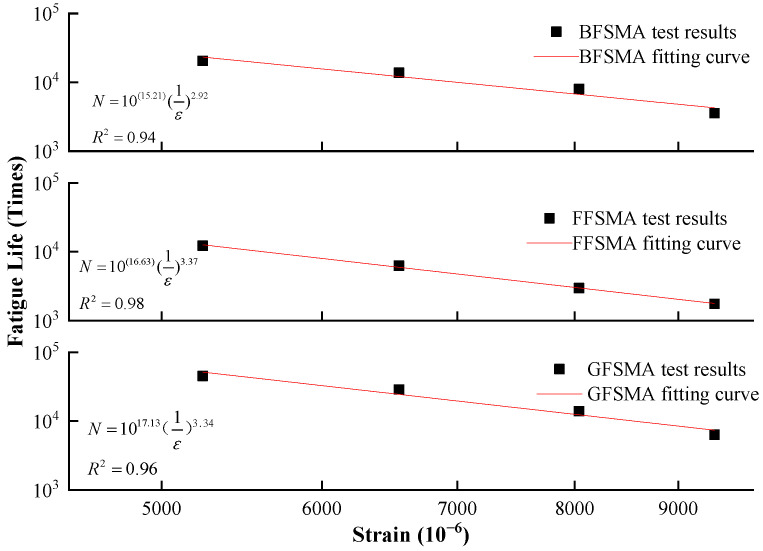
Fitting between strain level and fatigue life.

**Figure 14 materials-16-00822-f014:**
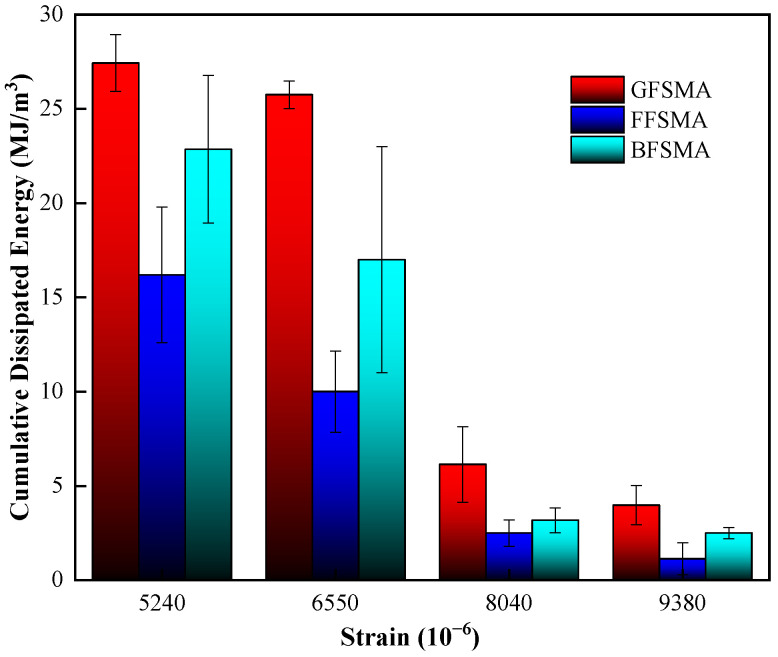
Cumulative dissipated energy results of mixtures.

**Figure 15 materials-16-00822-f015:**
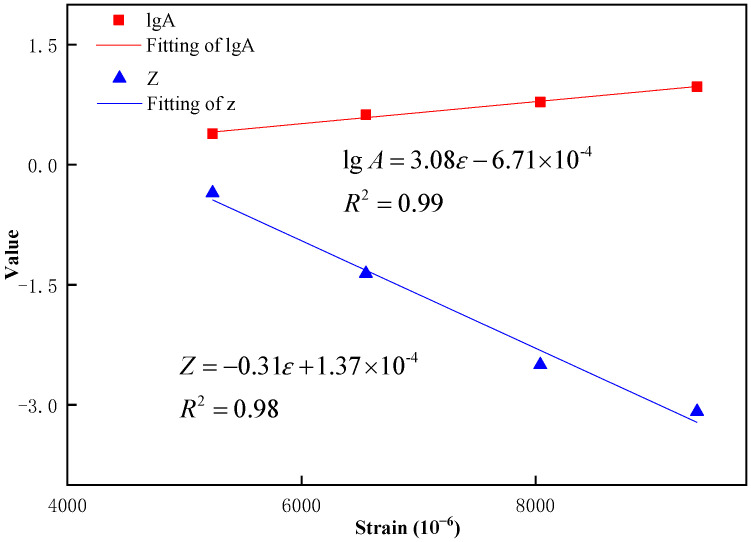
Fitting results of parameters.

**Figure 16 materials-16-00822-f016:**
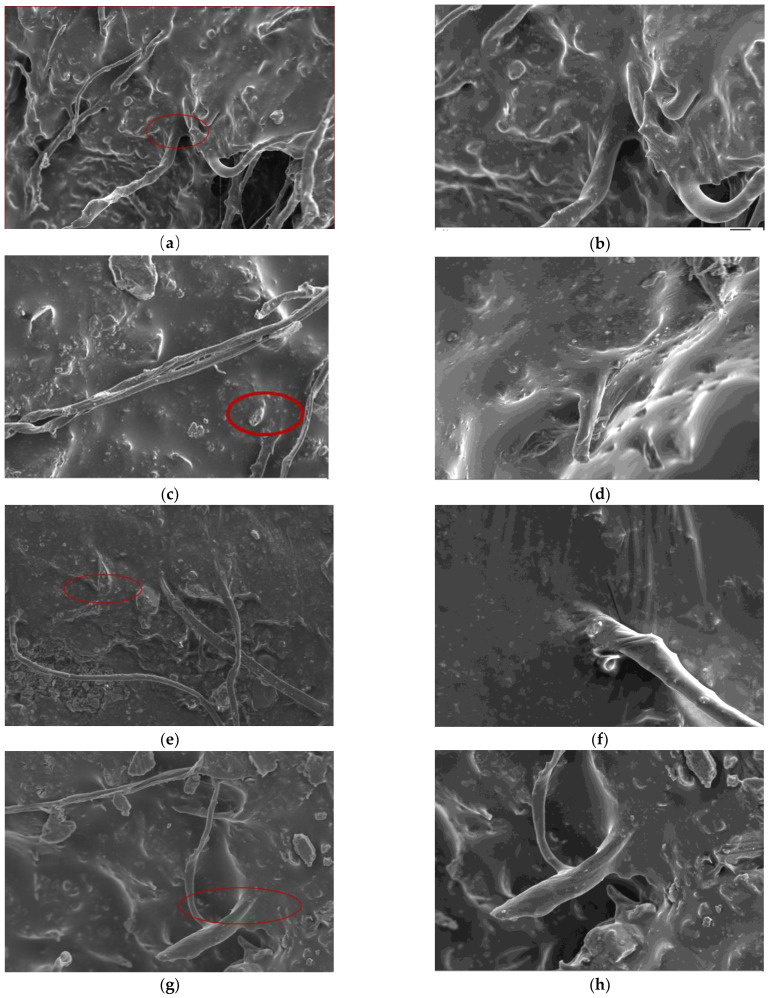
Microscopic Morphology: (**a**) 0.30 wt% GFSMA; (**b**) local zoom of 0.30 wt% GFSMA; (**c**) 0.40 wt% GFSMA; (**d**) local zoom of 0.40 wt% GFSMA; (**e**) 0.45 wt% GFSMA; (**f**) local zoom of 0.45 wt% GFSMA; (**g**) 0.5 wt% GFSMA; (**h**) local zoom of 0.5 wt% GFSMA; (**i**) 0.55 wt% GFSMA; (**j**) local zoom of 0.55 wt% GFSMA; (**k**) 0.45 wt% FFSMA; (**l**) local zoom of 0.45 wt% FFSMA; (**m**) 0.50 wt% BFSMA; (**n**) local zoom of 0.50 wt% BFSMA.

**Table 1 materials-16-00822-t001:** Basic performance index test results of the SBS-modified asphalt.

Index	Unit	Result	Method
Penetration (25 °C, 100 g, 5 s)	0.1 mm	43.3	ASTM D5-13
Ductility (5 °C, 5 cm/min)	cm	26.4	ASTM D113-17b
Softening point	°C	91.4	ASTM D36-14
Flash point	°C	>230	ASTM D92-12b
Density	g/cm^3^	1.042	ASTM D70-17a
**Residue after TFOT**			
Mass loss	%	0.3	ASTM D6-11
Penetration ratio (25 °C)	%	74.3	ASTM D5-13
Ductility (5 °C, 5 cm/min)	cm	22	ASTM D113-17b

**Table 2 materials-16-00822-t002:** Performance of fibers (offered by manufacturers).

		Fiber Type		
Index	Unit	Granular Lignin	Flocculent Lignin	Basalt
Length	mm	1.5	1	6
Diameter	mm	8	7	17
Oil absorption	times	6.09	7.14	3.20
pH value	/	7.40	7.30	7.60
Tensile strength	MPa	<300	<300	>2000
Density	g/cm^3^	1.15	0.91	2.75

**Table 3 materials-16-00822-t003:** Technical index test results of aggregates.

		Aggregate			
Index	Unit	Coarse	Fine	Filler	Method
Specific gravity	g/cm^3^	2.756			ASTM C127-15
Specific gravity	g/cm^3^		2.722		ASTM C128-15
Specific gravity	g/cm^3^			2.715	ASTM D854-14
Los Angeles abrasion	%	11.6			ASTM C131-14
Flat and elongated particles	%	1.6			ASTM D4791-19
Fine aggregate angularity	%		37		AASHTO T304-17

**Table 4 materials-16-00822-t004:** SMA13 gradation.

Sieve Size/mm	16	13.2	9.5	4.75	2.36	1.18	0.6	0.3	0.15	0.075
Upper limit/%	100	100	75	34	26	24	20	16	15	12
Lower limit/%	100	90	50	20	15	14	12	10	9	8
Target/%	100	95	61	24.4	21	17.8	15.5	13.5	11.7	10.2

**Table 5 materials-16-00822-t005:** Result of OAC.

Fiber Dosage (wt%)		OAC (%)	
	GFSMA	FFSMA	BFSMA
0.30	5.75	6.00	-
0.40	5.80	6.08	5.50
0.45	5.85	6.11	5.54
0.50	5.90	6.15	5.60
0.55	5.94	6.17	5.63

**Table 6 materials-16-00822-t006:** Result of fatigue life.

Strain	Specimen Number	GFSMA		FFSMA		BFSMA	
		Fatigue Life (Times)	Average (Times)	Fatigue Life (Times)	Average (Times)	Fatigue Life (Times)	Average (Times)
5048	1	48,824	45,209	11,654	12,200	23,161	20,513
	2	42,361		13,502		17,238	
	3	43,894		12,404		17,921	
	4	46,267		13,795		18,286	
	5	44,699		9644		15,960	
6310	1	28,708	28,671	6481	6273	15,478	13,755
	2	27,491		5301		11,117	
	3	29,905		7499		13,873	
	4	31,250		6158		10,990	
	5	26,000		5925		17,316	
7572	1	12,452	13,950	2630	2954	8165	7990
	2	13,500		3307		10,940	
	3	17,018		3357		6771	
	4	11,617		2562		9804	
	5	15,164		2914		4269	
8834	1	6836	6339	2125	1742	3404	3555
	2	5764		1345		1847	
	3	6375		1319		3163	
	4	5922		1844		5085	
	5	6757		2077		4280	

**Table 7 materials-16-00822-t007:** Fitting Results of Parameters.

Strain (10^−6^)	lgA	Z	$R^{2}$
5240	−0.35	0.39	0.94
6550	−1.36	0.62	0.99
8040	−2.49	0.78	0.96
9380	−3.08	0.97	0.99

## Data Availability

Not applicable.

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
