# Peer review of "Influence of Fiber Type and Dosage on Tensile Property of Asphalt Mixture Using Direct Tensile Test"

_materials, 2023, doi:10.3390/ma16020822_

Round 1

Reviewer 1 Report

The investigation is carefully designed, implemented and reported.

The Authors discuss “fracture toughness” in the context of microscopic observations of the failure zone. However, it appears that the notation here possibly does not correspond to the Fracture Mechanics meaning. This reviewer would welcome some discussion of Fracture Mechanics. This reviewer also would discuss welcome a somewhat more complete mechanical characterization, including stiffness properties.

This reviewer does not think “tensile fatigue” is a property that shall be improved.

It might be important to know the properties of the asphalt without fiber reinforcement.

The fitted polynomial functions appear clarifying. This reviewer also would be interested in stress-strain – curves.

Fatigue strain levels are given in units unknown to this reviewer. The reviewer was thinking strains are dimesionless. It might be important to clarify whether some irrecoverable deformation appeared during the strain-controlled tensile fatigue test.

Author Response

Response to Reviewer 1 Comments

Dear section editor Evangelos Bellos and reviewers,

Thank you for your letter and for the reviewers’ comments on our manuscript entitled “Influence of Fiber Type and Dosage on Tensile Property of Asphalt Mixture Using Direct Tensile Test”(materials-2121783). Those comments are not only valuable guidance for revising and improving our paper, but also important reference to our researches. We have studied the comments carefully and have made corrections which we hope to meet your approval requirements. Revised portion are marked in red in the revised manuscript attachment uploaded. The main corrections in paper and responses to the reviewers’ comments are as follows:

The investigation is carefully designed, implemented and reported.

  1. The Authors discuss “fracture toughness” in the context of microscopic observations of the failure zone. However, it appears that the notation here possibly does not correspond to the Fracture Mechanics meaning. This reviewer would welcome some discussion of Fracture Mechanics. This reviewer also would discuss welcome a somewhat more complete mechanical characterization, including stiffness properties.

Response: The authors appreciate the reviewer’s comment. Relevant discussion about fracture mechanics has been added in the revised manuscript (Line343-350). And discussion on stiffness modulus has also been added in the revised manuscript (Line461-505).  

  1. This reviewer does not think “tensile fatigue” is a property that shall be improved.

Response: The authors appreciate the reviewer’s comment. Asphalt mixture pavement is exposed to the repeated action of vehicle loading and temperature loading inevitably in service period, which can also result in fatigue cracking distress, even though the present tensile strain is lower than the tensile capacity of mixture itself. From this point of view, study on the tensile fatigue performance of asphalt mixture is necessary.

  1. It might be important to know the properties of the asphalt without fiber reinforcement.

Response: The authors appreciate the reviewer’s comment. This point is indeed worthy of further research. As mentioned above, we did not conduct test on blank control group in this study. The reason is as follows A large amount of asphalt demand is a classic feature of SMA (Stone Mastic Asphalt) mixture, which can inevitably result in leakage and flow phenomena, in order to overcome this difficult, pioneers added fiber into the mixture and then fiber has been a necessary component of SMA.

  1. The fitted polynomial functions appear clarifying. This reviewer also would be interested in stress-strain – curves.

Response: The authors appreciate the reviewer’s comment. We have added the stress-strain curves in the revised manuscript and given detailed illustration (Line270-310).

  1. Fatigue strain levels are given in units unknown to this reviewer. The reviewer was thinking strains are dimensionless. It might be important to clarify whether some irrecoverable deformation appeared during the strain-controlled tensile fatigue test.

Response: Thank you for reminding us. We have modified the terminology from “strain level” to “strain “(Line 580,586,605,607,613, 616-618,632-633). And in the fatigue test, there did appear irrecoverable defomation.

Thank you again for the kind advice. We tried our best to improve the manuscript and made some changes in the manuscript. And the changes were marked in red in the revised paper. We appreciate for reviewers' warm work earnestly, and hope that the correction will meet with the approval.

Once again, thank you very much for your comments and suggestions adn wish you a happy new year!

Kind regards

The authors.

Shuyao Yang, Zhigang Zhou and Kai Li

Reviewer 2 Report

1. You mention that - "many defects have occurred on asphalt concrete pavement resulting from tensile performance deficiency, among which, cracking is the most typical one." - mention any reference/justification on how cracking is related to tensile performance deficiency.

2. Explain what is "RTFOT" ?

3. Did you conduct the tests to obtain performance indices for SBS modified asphalt shown in Table 1? If so, briefly explain the test procedures. Same goes for Table 3.

4. How did you obain the Optimum Asphalt Content (OAC)? Explain the method/formula used. Also elaborate the results shown in Table 5. Why is there no entry for BFSMA at 0.3% fiber dosage?

5. Why does the tensile strength drop after a particular value of fiber dosage? Could you shed light on mechanics of interaction between the matrix and fiber phases which results in this behavior?

6. In case of ultimate strain as well, add discussion on possible reasons for saturation in ultimate strain with increase in fiber dosage.

7. Similar to observation for tensile strength, provide some explanation for the reduction of strain energy density after a particular value of fiber content. Simply reporting the results does not provide much insights into the mechanics of fiber dosage to asphalt.

8. How was the optimum dosage determined to maximize all these parameters (tensile strength, ultimate strain, strain energy density)? Any optimization methodology? Also, mention the optimum fiber content.

9. No need to mention lack of space in the manuscript. It gives a bad impression implying that the content covered in the paper is inadequate by itself. 

10. It would be better to add a figure of stress-strain relationships obtained from the tensile tests to get an idea of the mechanical behavior of the developed materials.

11. A general remark - check for English language typing and grammatical errors throughout the manuscript.

Author Response

Response to Reviewer 2 Comments

Dear section editor Evangelos Bellos and reviewers,

Thank you for your letter and for the reviewers’ comments on our manuscript entitled “Influence of Fiber Type and Dosage on Tensile Property of Asphalt Mixture Using Direct Tensile Test”(materials-2121783). Those comments are not only valuable guidance for revising and improving our paper, but also important reference to our researches. We have studied the comments carefully and have made corrections which we hope to meet your approval requirements. Revised portion are marked in red in the revised paper. The main corrections in paper and responses to the reviewers’ comments are as follows:

  1. You mention that - "many defects have occurred on asphalt concrete pavement resulting from tensile performance deficiency, among which, cracking is the most typical one." - mention any reference/justification on how cracking is related to tensile performance deficiency.

Response: The authors appreciate the reviewer’s comment and are very sorry for our mistake in reference citation. We have added the relative literatures citations in the revised manuscript(Line53).

AE Álvarez; Walubita, L. F.; F Sánchez. Using fracture energy to characterize the hot mix asphalt cracking resistance based on the direct- tensile test. Rev. Fac. Ing. 2012, 20, 126-137.

Xu, W.Z. Research on cracking behavior of high dosage hot reclaimed asphalt mixture based on digital speckle method. Master degree, Shenyang university of civil engineering and architecture, China, 2021.

  1. Explain what is "RTFOT" ?

Response: The authors appreciate the reviewer’s comment. We apologize for our spelling mistake and in complete statement on terminology. We have corrected “RTFOT” into “TFOT” (Line 142 ). “TFOT” is the abbreviation of “Thin Film Oven Test” and is a way to simulating the short term aging of asphalt. The procedure is displayed in the next response.

  1. Did you conduct the tests to obtain performance indices for SBS modified asphalt shown in Table 1? If so, briefly explain the test procedures. Same goes for Table 3.

Response: The authors appreciate the reviewer’s comment. We conducted the mentioned tests and relative test procedures are explained in the following tables.

Table a. Procedure explanation for Table1

Index

Procedure

Penetration

(1)        Keep vessel containing asphalt specimen in thermostatic water for at least 1.5h.

(2)        Move the vessel that has reached the constant temperature and transfer it to the three-legged bracket in a flat-bottomed glass vessel.

(3)        Slowly lower the pin linkage. Reset the displacement meter to zero when the needle is in contact with the surface of the specimen.

(4)        Start the test, press the release button, Record the value after the needle automatically stop falling.

Ductility

(1)    Specimen molding and insulation in the specific temperature.

(2)    Take the insulated test piece and the base plate into the water tank of the ductility tester.

(3)    Move away the base plate, put the holes at both ends of the mold on the corresponding metal column and then remove the side mold.

(4)    Start the tester and record the value when the specimen breaks,

Softening point

(1)    Inject pre-heated glycerin into the beaker.

(2)    Take out the specimen ring containing asphalt from the thermostatic tank and put the whole ring frame into the beaker after assembling.

(3)    Move the beaker on the heating furnace with asbestos mesh, and then adjust the steel ball to be in the middle of the sample.

(4)    Start the electromagnetic oscillation agitator. Record the temperature  value as soon as the asphalt contacts with the surface of the lower floor.

Flash point

(1)       Heat the sample.

(2)       Sweep the test flame of the igniter horizontally along the center of the with a radius of 150mm every 2℃ when the sample temperature is lower than the predicted flash point by 28℃.

(3)       Record the temperature at which the blue flame fistly appears on the liquid surface.

Density

mass of gravity bottle, g;

—— mass of gravity bottle and asphalt, g;

——, g

density of water, g/cm3

(1)   Wash the gravity bottle, dry and then weight it as  .

(2)   Put the beaker containing distilled water into the constant temperature tank and then put the above bottle into the beaker. The beaker containing distilled water is put into the constant temperature tank.

(3)   Put the bottle and its cap into the beaker.

(4)   After the water temperature in the beaker reached the specific temperature, keep it for 30min so that bubbles in the bottle rise to the water surface. Plug the cap the bottle tightly after confirming the bottle has been kept at a constant temperature without bubbles.

(5)   Take out the bottle and wipe the water on it and weigh it as .

(6)   Fill the gravity bottle with asphalt sample

(7)   Move the gravity bottle to the dryer and the bottle is weighed with cap together after being cooling for no less than an one hour (.

(8)   Repeat(2)(3)(4).

(9)   Take out the bottle and wipe the water on it and weigh it .

TFOT

(1)    Place asphalt sample of the water-free material to be tested in tared container. and weigh as .

(2)    Bring the oven to the temperature of 163℃ and then place the container and the weighed asphalt on and near the circumference of the circular shelf.

(3)    Close the oven and rotate the shelf at a rate of 5rpm to 6rpm.

(4)    Maintain the temperature at 163 for 5h and in no case shall the time that a sample is in the oven be more than 5.25h.

(5)    Remove the sample from the oven, cool to room temperature.

Mass loss( after TFOT)

 ——mass loss of sample after TFOT, %;

 ——mass of container, g;

 ——mass of container and sample before TFOT, g;

 ——mass of container and sample after TFOT.

(1)    weigh the sample in TFOT procedure (5) as

(2)    Calculate using the equation displayed in the left column.

Penetration ratio( after TFOT)

 ——penetration ratio of sample after TFOT, %;

 ——penetration before TFOT, 0.1mm;

 ——penetration after TFOT, 0.1mm.

(1)    Take the sample afte TFOT into a certain container and heat the sample on the heating furnace, stir properly to fully make it to flow state;

(2)    Put the flow asphalt into the penetration mold and operation as the penetration test procedure mentioned above.

(2)   Calculate using the equation displayed in the left column.

Ductility( after TFOT)

(1)    Operate as step (1) of penetration ratio (after TFOT).

(2)    Put the flow asphalt into the ductility mold and operation as the ductility test procedure mentioned above.

Table b. Procedure explanation for Table3

Index

Procedure

Specific gravity (coarse aggregate)

——mass of aggregates after drying, g;

——mass of aggregates in water, g;

—— Specific gravity.

(1)    Submerge aggregates into water for 24h.

(2)    Put aggregates into the basket of the immersion balance, record the value on the balance (.

(3)    Remove aggregates from the basket and absorb the water of the aggregates by wet towel or pure-cotton cloth till the aggregate reach surface drying state.

(4)    Put theses aggregates on a dish, dry them in an oven, cool to room temperature, and weigh (.

Specific gravity(fine aggregate)

—— mass of aggregates after drying, g;

—— mass of water and volumetric flask,g

—— mass of water, volumetric flask and sample, g;

—— Specific gravity.

(1)    Put 300g ()dried sample into a volumetric flask with water half- filled.

(2)    Shake the flask to eliminate bubbles, plug the cork tightly,  keep it static for 24h, add water using a dropper till the liquid level reached the tick mark, plug the cork tightly, dry, weigh .

(3)    Empty the flask, wash the flask, inject water to the flask, plug the cork tightly, dry , weigh.

Specific gravity(filler)

——total mass of spoon, funnel, container and filler before test, g;

—— total mass of spoon, funnel, container and filler after test, g;

——scale value of pycnometer before adding filler, Ml;

—— scale value of pycnometer after adding filler, Ml;

——density of water at test temperature, g/cm3;

—— Specific gravity.

(1)    Put filler in container, dry in oven, cool.

(2)    Weigh the container filled with filler, spoon and funnel ().

(3)    Put the pycnometer containing distilled water(≤1mL) into a constant-temperature-water tank for at least 2h, then record the scale value ().

(4)    Gradually add filler into the pycnometer using spoon till the liquid level is close to the maximum scale. And shake it lightly to emit the air in it.

(5)    Again put the pycnometer into the water tank. Record the scale when the temperature of the pycnometer keep stable().

(6)    Weigh the container, the remaining filler, spoon and funnel().

Los Angeles abrasion

 ——Los Angeles abrasion loss, %;

 ——mass of aggregates filled in the roller, g;

 ——mass of aggregates after sieving, washing and drying, g.

(1)    Clean aggregates , dry and then weigh as .

(2)    Put the aggregates into roller and add the steel ball into the roller and start the tester.

(3)    Remove the steel ball and aggregates from the roller after rotating for the specific times.

(4)    Remove the fine debris using sieve with 1.77mm square hole, weigh the aggregates remained on the sieve after washing and drying .

Flat and elongated particles

——content of flat and elongated particles

, %;

——mass of aggregate used in the test, g;

——mass of flat and elongated particles, g.

(1)    Sieve aggregates through the standard 4.75mm sieve and weigh as .

(2)    Lay the aggregates horizontally on the table, select aggregates with cubic shape by visual observation and remove them.

(3)    Measure the of the maximum horizontal length and the maximum height of the side of remaining aggregates laid on the table one by one. Then select aggregates of which the maximum horizontal length is more than 3 times the maximum height of the side as the falt and elongated particles and weigh as .

Fine aggregate angularity

——fine aggregate angularity, %;

——mass of the receiving container free of water, g;

—— mass of the receiving container filled with water, g;

——mass of the receiving container with aggregates, g;

—— bulk volume relative density of fine aggregate.

(1)    Weigh the receiving container free of water .

(2)    Fill the receiving container with water and weigh .

(3)    Sieve, wash and dry the aggregates.

(4)    Assemble the cylinder container, funnel and into a whole.

(5)    Gradually pour aggregates into the assembled structure and then aggregates drop into the receiving container.

(6)    Weigh the receiving container with aggregates in it .

  1. How did you obtain the Optimum Asphalt Content (OAC)? Explain the method/formula used. Also elaborate the results shown in Table 5. Why is there no entry for BFSMA at 0.3% fiber dosage?

Response: The authors appreciate the reviewer’s comment.

  1. The acquisition of OAC was followed by the following procedure.
  • Set a estimation value of asphalt content according to the relevant calculation method in “Technical Specification for Highway Asphalt Pavement Construction”(JTG F40-2004).
  • Take the estimated value as the median and set five different asphalt content values at a certain interval.
  • Execute marshall test respectively on asphalt mixtures under the five asphalt content values mentioned above.
  • Plot the curves of marshall test indexes (bulk density, void ratio, marshall stability, flow value, void in mineral aggregate, void filled with asphalt) with the variation of asphalt content value.
  • Find the maximum bulk density, maximum marshall stability, medium void ratio and medium void in mineral aggregate, and their corresponding asphalt content values were set as a1,a2,a3 and a
  • Calculate:
  • Acquire the asphalt content region in which the six marshal test indexes mentioned above were all meet the requirement by standard and then set the maximum and minimum asphalt content value according to the region espectively as OACmax and OAC
  • Calculate:
  • The optimum asphalt content was acquired by the following:
  • Operate leakage test and Cantabro test on asphalt mixture with the optimum asphalt content to demonstrate whether this asphalt content meet the requirement of leakage and dispersion.
  • After obtaining the optimum amount of asphalt, it is necessary to check again whether the optimum amount of asphalt conforms to the requirements of leakage and flight dispersion.
  • corresponding asphalt content values at which.
  1. Detailed elaboration issue has been added to the revised manuscript (Line 173-183).
  2. It is stipulated in Technical Specification for Construction of Highway Asphalt Pavements (JTG F40-2004) that the dosage of lignin fiber should not be less than 0.30wt% of the total weight of the mixture, and the dosage of basalt fiber should not be less than 0.4 wt%. So, the lowest fiber dosage of BFSMA was set as 0.40wt% and there was no entry for BFSMA at 0.3wt% fiber dosage.
  3. Why does the tensile strength drop after a particular value of fiber dosage? Could you shed light on mechanics of interaction between the matrix and fiber phases which results in this behavior?

Response: The authors appreciate the reviewer’s comment. We have added explanation (Line 362-371) as follows: An appropriate elevation of fiber dosage, the average fiber distance could be shortened so that there could be more fibers participating in the adhesion with matrix, which would strengthen the interface between the matrix and fiber and eventually improve the tensile strength of mixture. However, the continuous rising of fiber dosage brought about the enlargement of coverage of fiber and then the narrowness of the effective bonding interface between fiber and asphalt, which could lead to the decline of tensile strength of mixture instead.

  1. In case of ultimate strain as well, add discussion on possible reasons for saturation in ultimate strain with increase in fiber dosage.

Response: The authors appreciate the reviewer’s comment. We have added explanation in the revised manuscript (Line 415-423)

  1. Similar to observation for tensile strength, provide some explanation for the reduction of strain energy density after a particular value of fiber content. Simply reporting the results does not provide much insights into the mechanics of fiber dosage to asphalt.

Response: The authors appreciate the reviewer’s comment. We have added relative illustration (Line 524-535).

  1. How was the optimum dosage determined to maximize all these parameters (tensile strength, ultimate strain, strain energy density)? Any optimization methodology? Also, mention the optimum fiber content.

Response: Thank you for your advice. This point is indeed worthy of further research. As mentioned above, we did not detemine the optimum dosage to maxmize all indexes. But in the paper, we discussed the variation regulation of indexes(tensile strength, ultimate strain and stain energy) with the variation of fiber dosage in monotonic tensile test and selected the fiber dosage at which the strain enegy density reached the peak value as the optimum fiber dosage respectively for three kinds of fiber reinforced asphalt mixture.The reason for choosing the strain enegy density as the key index to select the optimum fiber dosage is as follows:On one hand, the variation regulation for the tensile strength and the ultimate strain with the variation of fiber dosage for each kind of fiber-reinforced asphalt mixture showed great difference and And the two indexes reached their own peak value at different fiber dosage, which could subsequently result in an inconsistency between the optimum fiber dosage determined respectively based on the two indicators. On the other hand, strain energy density is a comprehensive indicator which can reflect both the tensile strength and ultimate strain[35]. Moreover, for each kind of fiber reinforced mixture at thei own optimum fiber dosage selected by this method, both the tensile stength and ultimate strain were in a satisfying level.

  1. No need to mention lack of space in the manuscript. It gives a bad impression implying that the content covered in the paper is inadequate by itself. 

Response:  Thank you for your advice. I have deleted the relative sentences.

  1. It would be better to add a figure of stress-strain relationships obtained from the tensile tests to get an idea of the mechanical behavior of the developed materials.

Response: The authors appreciate the reviewer’s comment. We have added the stress-strain curves in the revised manuscript and given detailed illustration (Line270-310).

  1. A general remark - check for English language typing and grammatical errors throughout the manuscript.

Response: The authors appreciate the reviewer’s comment. We have checked and conduct modification in the revised manuscript.

Thank you again for the kind advice. We tried our best to improve the manuscript and made some changes in the manuscript. And the changes were marked in red in the revised paper. We appreciate for reviewers' warm work earnestly, and hope that the correction will meet with the approval.

Once again, thank you very much for your comments and suggestions adn wish you a happy new year!

Kind regards

The authors.

Shuyao Yang, Zhigang Zhou and Kai Li

Response to Reviewer 2 Comments

Dear section editor Evangelos Bellos and reviewers,

Thank you for your letter and for the reviewers’ comments on our manuscript entitled “Influence of Fiber Type and Dosage on Tensile Property of Asphalt Mixture Using Direct Tensile Test”(materials-2121783). Those comments are not only valuable guidance for revising and improving our paper, but also important reference to our researches. We have studied the comments carefully and have made corrections which we hope to meet your approval requirements. Revised portion are marked in red in the revised paper. The main corrections in paper and responses to the reviewers’ comments are as follows:

  1. You mention that - "many defects have occurred on asphalt concrete pavement resulting from tensile performance deficiency, among which, cracking is the most typical one." - mention any reference/justification on how cracking is related to tensile performance deficiency.

Response: The authors appreciate the reviewer’s comment and are very sorry for our mistake in reference citation. We have added the relative literatures citations in the revised manuscript(Line53)..

AE Álvarez; Walubita, L. F.; F Sánchez. Using fracture energy to characterize the hot mix asphalt cracking resistance based on the direct- tensile test. Rev. Fac. Ing. 2012, 20, 126-137.

Xu, W.Z. Research on cracking behavior of high dosage hot reclaimed asphalt mixture based on digital speckle method. Master degree, Shenyang university of civil engineering and architecture, China, 2021.

  1. Explain what is "RTFOT" ?

Response: The authors appreciate the reviewer’s comment. We apologize for our spelling mistake and in complete statement on terminology. We have corrected “RTFOT” into “TFOT” (Line 142 ). “TFOT” is the abbreviation of “Thin Film Oven Test” and is a way to simulating the short term aging of asphalt. The procedure is displayed in the next response.

  1. Did you conduct the tests to obtain performance indices for SBS modified asphalt shown in Table 1? If so, briefly explain the test procedures. Same goes for Table 3.

Response: The authors appreciate the reviewer’s comment. We conducted the mentioned tests and relative test procedures are explained in the following tables.

Table a. Procedure explanation for Table1

Index

Procedure

Penetration

(1)        Keep vessel containing asphalt specimen in thermostatic water for at least 1.5h.

(2)        Move the vessel that has reached the constant temperature and transfer it to the three-legged bracket in a flat-bottomed glass vessel.

(3)        Slowly lower the pin linkage. Reset the displacement meter to zero when the needle is in contact with the surface of the specimen.

(4)        Start the test, press the release button, Record the value after the needle automatically stop falling.

Ductility

(1)    Specimen molding and insulation in the specific temperature.

(2)    Take the insulated test piece and the base plate into the water tank of the ductility tester.

(3)    Move away the base plate, put the holes at both ends of the mold on the corresponding metal column and then remove the side mold.

(4)    Start the tester and record the value when the specimen breaks,

Softening point

(1)    Inject pre-heated glycerin into the beaker.

(2)    Take out the specimen ring containing asphalt from the thermostatic tank and put the whole ring frame into the beaker after assembling.

(3)    Move the beaker on the heating furnace with asbestos mesh, and then adjust the steel ball to be in the middle of the sample.

(4)    Start the electromagnetic oscillation agitator. Record the temperature  value as soon as the asphalt contacts with the surface of the lower floor.

Flash point

(1)       Heat the sample.

(2)       Sweep the test flame of the igniter horizontally along the center of the with a radius of 150mm every 2℃ when the sample temperature is lower than the predicted flash point by 28℃.

(3)       Record the temperature at which the blue flame fistly appears on the liquid surface.

Density

mass of gravity bottle, g;

—— mass of gravity bottle and asphalt, g;

——, g

density of water, g/cm3

(1)   Wash the gravity bottle, dry and then weight it as  .

(2)   Put the beaker containing distilled water into the constant temperature tank and then put the above bottle into the beaker. The beaker containing distilled water is put into the constant temperature tank.

(3)   Put the bottle and its cap into the beaker.

(4)   After the water temperature in the beaker reached the specific temperature, keep it for 30min so that bubbles in the bottle rise to the water surface. Plug the cap the bottle tightly after confirming the bottle has been kept at a constant temperature without bubbles.

(5)   Take out the bottle and wipe the water on it and weigh it as .

(6)   Fill the gravity bottle with asphalt sample

(7)   Move the gravity bottle to the dryer and the bottle is weighed with cap together after being cooling for no less than an one hour (.

(8)   Repeat(2)(3)(4).

(9)   Take out the bottle and wipe the water on it and weigh it .

TFOT

(1)    Place asphalt sample of the water-free material to be tested in tared container. and weigh as .

(2)    Bring the oven to the temperature of 163℃ and then place the container and the weighed asphalt on and near the circumference of the circular shelf.

(3)    Close the oven and rotate the shelf at a rate of 5rpm to 6rpm.

(4)    Maintain the temperature at 163 for 5h and in no case shall the time that a sample is in the oven be more than 5.25h.

(5)    Remove the sample from the oven, cool to room temperature.

Mass loss( after TFOT)

 ——mass loss of sample after TFOT, %;

 ——mass of container, g;

 ——mass of container and sample before TFOT, g;

 ——mass of container and sample after TFOT.

(1)    weigh the sample in TFOT procedure (5) as

(2)    Calculate using the equation displayed in the left column.

Penetration ratio( after TFOT)

 ——penetration ratio of sample after TFOT, %;

 ——penetration before TFOT, 0.1mm;

 ——penetration after TFOT, 0.1mm.

(1)    Take the sample afte TFOT into a certain container and heat the sample on the heating furnace, stir properly to fully make it to flow state;

(2)    Put the flow asphalt into the penetration mold and operation as the penetration test procedure mentioned above.

(2)   Calculate using the equation displayed in the left column.

Ductility( after TFOT)

(1)    Operate as step (1) of penetration ratio (after TFOT).

(2)    Put the flow asphalt into the ductility mold and operation as the ductility test procedure mentioned above.

Table b. Procedure explanation for Table3

Index

Procedure

Specific gravity (coarse aggregate)

——mass of aggregates after drying, g;

——mass of aggregates in water, g;

—— Specific gravity.

(1)    Submerge aggregates into water for 24h.

(2)    Put aggregates into the basket of the immersion balance, record the value on the balance (.

(3)    Remove aggregates from the basket and absorb the water of the aggregates by wet towel or pure-cotton cloth till the aggregate reach surface drying state.

(4)    Put theses aggregates on a dish, dry them in an oven, cool to room temperature, and weigh (.

Specific gravity(fine aggregate)

—— mass of aggregates after drying, g;

—— mass of water and volumetric flask,g

—— mass of water, volumetric flask and sample, g;

—— Specific gravity.

(1)    Put 300g ()dried sample into a volumetric flask with water half- filled.

(2)    Shake the flask to eliminate bubbles, plug the cork tightly,  keep it static for 24h, add water using a dropper till the liquid level reached the tick mark, plug the cork tightly, dry, weigh .

(3)    Empty the flask, wash the flask, inject water to the flask, plug the cork tightly, dry , weigh.

Specific gravity(filler)

——total mass of spoon, funnel, container and filler before test, g;

—— total mass of spoon, funnel, container and filler after test, g;

——scale value of pycnometer before adding filler, Ml;

—— scale value of pycnometer after adding filler, Ml;

——density of water at test temperature, g/cm3;

—— Specific gravity.

(1)    Put filler in container, dry in oven, cool.

(2)    Weigh the container filled with filler, spoon and funnel ().

(3)    Put the pycnometer containing distilled water(≤1mL) into a constant-temperature-water tank for at least 2h, then record the scale value ().

(4)    Gradually add filler into the pycnometer using spoon till the liquid level is close to the maximum scale. And shake it lightly to emit the air in it.

(5)    Again put the pycnometer into the water tank. Record the scale when the temperature of the pycnometer keep stable().

(6)    Weigh the container, the remaining filler, spoon and funnel().

Los Angeles abrasion

 ——Los Angeles abrasion loss, %;

 ——mass of aggregates filled in the roller, g;

 ——mass of aggregates after sieving, washing and drying, g.

(1)    Clean aggregates , dry and then weigh as .

(2)    Put the aggregates into roller and add the steel ball into the roller and start the tester.

(3)    Remove the steel ball and aggregates from the roller after rotating for the specific times.

(4)    Remove the fine debris using sieve with 1.77mm square hole, weigh the aggregates remained on the sieve after washing and drying .

Flat and elongated particles

——content of flat and elongated particles

, %;

——mass of aggregate used in the test, g;

——mass of flat and elongated particles, g.

(1)    Sieve aggregates through the standard 4.75mm sieve and weigh as .

(2)    Lay the aggregates horizontally on the table, select aggregates with cubic shape by visual observation and remove them.

(3)    Measure the of the maximum horizontal length and the maximum height of the side of remaining aggregates laid on the table one by one. Then select aggregates of which the maximum horizontal length is more than 3 times the maximum height of the side as the falt and elongated particles and weigh as .

Fine aggregate angularity

——fine aggregate angularity, %;

——mass of the receiving container free of water, g;

—— mass of the receiving container filled with water, g;

——mass of the receiving container with aggregates, g;

—— bulk volume relative density of fine aggregate.

(1)    Weigh the receiving container free of water .

(2)    Fill the receiving container with water and weigh .

(3)    Sieve, wash and dry the aggregates.

(4)    Assemble the cylinder container, funnel and into a whole.

(5)    Gradually pour aggregates into the assembled structure and then aggregates drop into the receiving container.

(6)    Weigh the receiving container with aggregates in it .

  1. How did you obtain the Optimum Asphalt Content (OAC)? Explain the method/formula used. Also elaborate the results shown in Table 5. Why is there no entry for BFSMA at 0.3% fiber dosage?

Response: The authors appreciate the reviewer’s comment.

  1. The acquisition of OAC was followed by the following procedure.
  • Set a estimation value of asphalt content according to the relevant calculation method in “Technical Specification for Highway Asphalt Pavement Construction”(JTG F40-2004).
  • Take the estimated value as the median and set five different asphalt content values at a certain interval.
  • Execute marshall test respectively on asphalt mixtures under the five asphalt content values mentioned above.
  • Plot the curves of marshall test indexes (bulk density, void ratio, marshall stability, flow value, void in mineral aggregate, void filled with asphalt) with the variation of asphalt content value.
  • Find the maximum bulk density, maximum marshall stability, medium void ratio and medium void in mineral aggregate, and their corresponding asphalt content values were set as a1,a2,a3 and a
  • Calculate:
  • Acquire the asphalt content region in which the six marshal test indexes mentioned above were all meet the requirement by standard and then set the maximum and minimum asphalt content value according to the region espectively as OACmax and OAC
  • Calculate:
  • The optimum asphalt content was acquired by the following:
  • Operate leakage test and Cantabro test on asphalt mixture with the optimum asphalt content to demonstrate whether this asphalt content meet the requirement of leakage and dispersion.
  • After obtaining the optimum amount of asphalt, it is necessary to check again whether the optimum amount of asphalt conforms to the requirements of leakage and flight dispersion.
  • corresponding asphalt content values at which.
  1. Detailed elaboration issue has been added to the revised manuscript (Line 173-183).
  2. It is stipulated in Technical Specification for Construction of Highway Asphalt Pavements (JTG F40-2004) that the dosage of lignin fiber should not be less than 0.30wt% of the total weight of the mixture, and the dosage of basalt fiber should not be less than 0.4 wt%. So, the lowest fiber dosage of BFSMA was set as 0.40wt% and there was no entry for BFSMA at 0.3wt% fiber dosage.
  3. Why does the tensile strength drop after a particular value of fiber dosage? Could you shed light on mechanics of interaction between the matrix and fiber phases which results in this behavior?

Response: The authors appreciate the reviewer’s comment. We have added explanation (Line 362-371) as follows: An appropriate elevation of fiber dosage, the average fiber distance could be shortened so that there could be more fibers participating in the adhesion with matrix, which would strengthen the interface between the matrix and fiber and eventually improve the tensile strength of mixture. However, the continuous rising of fiber dosage brought about the enlargement of coverage of fiber and then the narrowness of the effective bonding interface between fiber and asphalt, which could lead to the decline of tensile strength of mixture instead.

  1. In case of ultimate strain as well, add discussion on possible reasons for saturation in ultimate strain with increase in fiber dosage.

Response: The authors appreciate the reviewer’s comment. We have added explanation in the revised manuscript (Line 415-423)

  1. Similar to observation for tensile strength, provide some explanation for the reduction of strain energy density after a particular value of fiber content. Simply reporting the results does not provide much insights into the mechanics of fiber dosage to asphalt.

Response: The authors appreciate the reviewer’s comment. We have added relative illustration (Line 524-535).

  1. How was the optimum dosage determined to maximize all these parameters (tensile strength, ultimate strain, strain energy density)? Any optimization methodology? Also, mention the optimum fiber content.

Response: Thank you for your advice. This point is indeed worthy of further research. As mentioned above, we did not detemine the optimum dosage to maxmize all indexes. But in the paper, we discussed the variation regulation of indexes(tensile strength, ultimate strain and stain energy) with the variation of fiber dosage in monotonic tensile test and selected the fiber dosage at which the strain enegy density reached the peak value as the optimum fiber dosage respectively for three kinds of fiber reinforced asphalt mixture.The reason for choosing the strain enegy density as the key index to select the optimum fiber dosage is as follows:On one hand, the variation regulation for the tensile strength and the ultimate strain with the variation of fiber dosage for each kind of fiber-reinforced asphalt mixture showed great difference and And the two indexes reached their own peak value at different fiber dosage, which could subsequently result in an inconsistency between the optimum fiber dosage determined respectively based on the two indicators. On the other hand, strain energy density is a comprehensive indicator which can reflect both the tensile strength and ultimate strain[35]. Moreover, for each kind of fiber reinforced mixture at thei own optimum fiber dosage selected by this method, both the tensile stength and ultimate strain were in a satisfying level.

  1. No need to mention lack of space in the manuscript. It gives a bad impression implying that the content covered in the paper is inadequate by itself. 

Response:  Thank you for your advice. I have deleted the relative sentences.

  1. It would be better to add a figure of stress-strain relationships obtained from the tensile tests to get an idea of the mechanical behavior of the developed materials.

Response: The authors appreciate the reviewer’s comment. We have added the stress-strain curves in the revised manuscript and given detailed illustration (Line270-310).

  1. A general remark - check for English language typing and grammatical errors throughout the manuscript.

Response: The authors appreciate the reviewer’s comment. We have checked and conduct modification in the revised manuscript.

Thank you again for the kind advice. We tried our best to improve the manuscript and made some changes in the manuscript. And the changes were marked in red in the revised paper. We appreciate for reviewers' warm work earnestly, and hope that the correction will meet with the approval.

Once again, thank you very much for your comments and suggestions adn wish you a happy new year!

Kind regards

The authors.

Shuyao Yang, Zhigang Zhou and Kai Li

Reviewer 3 Report

The article presents the influence of some fibre types and dosages on the tensile properties of asphalt mixtures using direct tensile tests. The results found optimal dosages depending on the type of fibre used. The article is well written and easily understandable. However, there are some important information that shall be clarified before publishing of the article:

1. Regarding fatigue tests, how many samples were tested of each mixture and level?

2. Figure 10 should present individual results not an average and error bars. What are these error bars?

3. Curve fitting in some figures show a cubic function fitted to 5 data points. This is meaningless, especially the R2 value. These shall be adjusted.

4. Results, such that of equation 6, should include upper and lower limits (say 90 or 95% confidence limits) because data scatter is significant.

Author Response

Response to Reviewer 3 Comments

Dear section editor Evangelos Bellos and reviewers,

Thank you for your letter and for the reviewers’ comments on our manuscript entitled “Influence of Fiber Type and Dosage on Tensile Property of Asphalt Mixture Using Direct Tensile Test”(materials-2121783). Those comments are not only valuable guidance for revising and improving our paper, but also important reference to our researches. We have studied the comments carefully and have made corrections which we hope to meet your approval requirements. Revised portion are marked in red in the revised paper. The main corrections in paper and responses to the reviewers’ comments are as follows:

The article presents the influence of some fibre types and dosages on the tensile properties of asphalt mixtures using direct tensile tests. The results found optimal dosages depending on the type of fibre used. The article is well written and easily understandable. However, there are some important information that shall be clarified before publishing of the article:

  1. Regarding fatigue tests, how many samples were tested of each mixture and level?

Response: The authors appreciate the reviewer’s comment. Five samples were tested of each mixture and level.

  1. Figure 10 should present individual results not an average and error bars. What are these error bars?

Response: The authors appreciate the reviewer’s comment. We have displayed the individual results in the revised manuscript(Table 6).

  1. Curve fitting in some figures show a cubic function fitted to 5 data points. This is meaningless, especially the R2 value. These shall be adjusted.

Response: The authors appreciate the reviewer’s comment. We have adjusted the fitting function in the revised manuscript (Fig 6).

  1. Results, such that of equation 6, should include upper and lower limits (say 90 or 95% confidence limits) because data scatter is significant.

Response: The authors appreciate the reviewer’s comment and thanks for reminding. We have added 95% confidence limits as a annotation in the fitting result(Line 644-645)..

Thank you again for the kind advice. We tried our best to improve the manuscript and made some changes in the manuscript. And the changes were marked in red in the revised paper. We appreciate for reviewers' warm work earnestly, and hope that the correction will meet with the approval.

Once again, thank you very much for your comments and suggestions adn wish you a happy new year!

Kind regards

The authors.

Shuyao Yang, Zhigang Zhou and Kai Li

Round 2

Reviewer 1 Report

Thank you for the opportunity to review this paper.

Reviewer 2 Report

The modifications made in the paper make it suitable for publication in Materials.

Reviewer 3 Report

Authors address the comments made by the reviewer.